# Following excited-state chemical shifts in molecular ultrafast x-ray photoelectron spectroscopy

D. Mayer [1,11], F. Lever [1,11], D. Picconi [2✉], J. Metje[1], S. Alisauskas [3], F. Calegari [4,5,6], S. Düsterer [3], C. Ehlert [7], R. Feifel[8], M. Niebuhr [1], B. Manschwetus [3], M. Kuhlmann[3], T. Mazza[9], M. S. Robinson[1,4,5], R. J. Squibb[8], A. Trabattoni [4], M. Wallner[8], P. Saalfrank[2], T. J. A. Wolf [10] & M. Gühr[1✉]

The conversion of photon energy into other energetic forms in molecules is accompanied by charge moving on ultrafast timescales. We directly observe the charge motion at a specific site in an electronically excited molecule using time-resolved x-ray photoelectron spectroscopy (TR-XPS). We extend the concept of static chemical shift from conventional XPS by the excited-state chemical shift (ESCS), which is connected to the charge in the framework of a potential model. This allows us to invert TR-XPS spectra to the dynamic charge at a specific atom. We demonstrate the power of TR-XPS by using sulphur $2p$-core-electron-emission probing to study the UV-excited dynamics of 2-thiouracil. The method allows us to discover that a major part of the population relaxes to the molecular ground state within 220–250 fs. In addition, a 250-fs oscillation, visible in the kinetic energy of the TR-XPS, reveals a coherent exchange of population among electronic states.

[1] Institut für Physik und Astronomie, Universität Potsdam, 14476 Potsdam, Germany. [2] Institut für Chemie, Universität Potsdam, 14476 Potsdam, Germany. [3] Deutsches Elektronen Synchrotron (DESY), 22607 Hamburg, Germany. [4] Center for Free-Electron Laser Science (CFEL), Deutsches Elektronen Synchrotron (DESY), 22607 Hamburg, Germany. [5] The Hamburg Centre for Ultrafast Imaging, Universität Hamburg, 22761 Hamburg, Germany. [6] Institut für Experimentalphysik, Universität Hamburg, 22761 Hamburg, Germany. [7] Heidelberg Institute for Theoretical Studies, HITS gGmbH, 69118 Heidelberg, Germany. [8] Department of Physics, University of Gothenburg, SE-41296 Gothenburg, Sweden. [9] European XFEL, 22869 Schenefeld, Germany. [10] Stanford PULSE Institute, SLAC National Accelerator Laboratory, Menlo Park, CA 94025, USA. [11] These authors contributed equally: D. Mayer, F. Lever. ✉email: david.picconi@uni-potsdam.de; mguehr@uni-potsdam.de

Light-induced charge flow in molecules forms the basis to convert photon energy into other energetic forms. The excitation of valence electrons by light triggers a change in charge density that eventually couples to nuclear motion. The complex interplay of nuclear and electronic degrees of freedom in the electronically excited states continues to move charge on an ultrafast time-scale by nonadiabatic couplings[1]. This in turn gives rise to phenomena like photoisomerization[2] and proton-coupled electron transfer[3] with rich applications in light-harvesting and photocatalysis[4]. Finding a way to image the charge flow in electronically excited systems, including organic molecules, on its natural timescale and with atomic precision would provide new ground for understanding molecular photophysics and excited-state (ES) reactivity.

X-ray photoelectron spectroscopy (XPS) is a proven tool to obtain information about local charge with atomic specificity in electronic ground states[5]. The tight localisation of core orbitals makes the method site selective. The ionisation potential (IP) measures the difference between neutral state and core-ionised state at a particular atom in a molecule. The so-called chemical shift (CS) of the IP reflects the charge at, and in close vicinity of, the probed atom. Within the potential approximation, the CS can be directly converted into local charge[6].

In this paper we generalise the CS concept known from conventional, static XPS to electronically *excited* states, introducing the excited-state chemical shift (ESCS, not to be confused with the ESCS in nuclear magnetic resonance). We test this on the thionucleobase 2-thiouracil (2-tUra), which is photoexcited to a $\pi\pi^*$ state by an ultraviolet (UV) light pulse (Fig. 1). The S 2p photoionization with a soft x-ray pulse, of photon energy $h\nu$,

leads to a photoelectron with kinetic energy $E_{kin} = h\nu - E_{bind}$. The molecular UV photoexcitation changes the local charge density at the probed atom (Fig. 1b bottom), which leads to a specific ESCS. We find a direct relation between the ESCS and the local charge at the probe site in analogy to the potential model for the CS in static XPS[6]. This allows one to circumvent complex calculations of IPs, while allowing for an interpretation based on chemical intuition. We show that the largest effect on the ESCS is due to electronic relaxation, especially if the local charge at the probed atom is grossly changed in the process. A smaller, but non-negligible effect, stems from geometry changes, which can also alter local charge at the probed atom.

Our studies extend existing theoretical[7–10] and experimental[11–13] time-resolved (TR)-XPS by the demonstration of direct local charge recovery from ESCS. The x-ray typical element- and site-specific responses[14,15] are also accomplished using the now well-established TR x-ray absorption spectroscopy (TR-XAS) method. In the soft x-ray range, TR-XAS has the capability to monitor electronic and nuclear dynamics[16–18], which has been demonstrated in ring-opening reactions[19], dissociation[20], intersystem-crossing[21], ionisation[22] as well as the interplay between $\pi\pi^*$ and $n\pi^*$ valence electronic states[23,24]. Hard x-ray absorption and emission spectroscopy is highly sensitive to charge and spin states, however, only on metal atoms within molecules[25,26]. The novel TR-XPS extends this characteristic to lighter atoms and has the advantage that a fixed x-ray wavelength can be used to address several elements and sites. In addition, the use for molecules in thick solvent jets can also be accomplished by employing harder x-rays for increased penetration depth of the radiation as well as escape depth of the photoelectrons for light-element XPS.

We demonstrate the opportunities provided by TR-XPS on electronically excited states of a thionucleobase. Photoexcited thionucleobases are interesting because of efficient relaxation into long-lived triplet states (see Ref. [27,28] and references therein) triggering applications as photoinduced cross-linkers[29,30] and photoinduced cancer therapy[28]. Among those, 2-tUra is one of the most-studied cases on an ultrafast scale[27], both experimentally, using transient absorption[31] and photoelectron spectroscopy[31–33], and theoretically in static calculations[34] and surface hopping trajectory simulations[35,36]. The latter predict coherent population exchange among electronic states. The model emerging from joint experimental-theoretical investigations includes ultrafast internal conversion from the photoexcited $S_2$ $\pi\pi^*$ state into the $S_1$ $n\pi^*$ state, followed by a sub-picosecond inter-system crossing[32]. Relaxation from the triplet states to the ground state has previously been observed with a time constant of several ten picoseconds[32] while also indirect evidence for an ultrafast direct ground-state (GS) relaxation from the photoexcited state has been reported[37].

In this work, we show that TR-XPS on electronically excited states allows us to identify the relaxation paths of 2-tUra by direct analysis of the ESCS and in addition comparisons to the calculated binding energy of different states and geometries. Most interestingly, we identify a ground state relaxation and the predicted coherent electronic population oscillation modulating the sulphur 2p binding energy periodically.

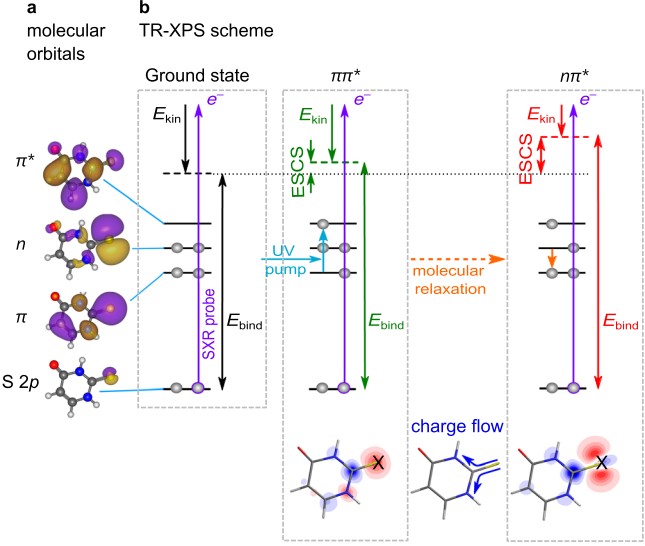

**Fig. 1 Schematic picture of TR-XPS and ESCS in 2-thiouracil in a molecular orbital representation. a** Molecular valence ($\pi$, $n$, $\pi^*$) orbitals and a core (sulphur 2p) orbital. **b** Probe of the S 2p core level with a binding energy $E_{bind}$ by means of a soft x-ray (SXR) light pulse, leading to a photoelectron with a kinetic energy, $E_{kin}$. A UV pump pulse (cyan arrow) excites the 2-thiouracil from its electronic ground state ($S_0$) to a $\pi\pi^*$ state ($S_2$) which then relaxes further, for instance into the $S_1$ ($n\pi^*$) state shown here. The difference in $E_{bind}$ with respect to the ground state is the excited-state chemical shift (ESCS $= E_{bind}^{excited\ state} - E_{bind}^{ground\ state}$). The molecular structures in the lower part of the panel represent the difference in charge density between the ground state and the respective excited states (red: decreased electron density, blue: increased electron density). Increase in positive charge at the sulphur site (marked with X) increases $E_{bind}$ and the ESCS.

## Results

**Difference spectra**. Figure 2a shows a photoelectron spectrum of 2-tUra obtained at the FLASH2 free-electron laser (FEL)[38] using a nominal photon energy of 272 eV and an average bandwidth of 1–2%. The electron spectra are taken with a magnetic-bottle electron spectrometer (MBES). We identify the sulphur 2p-photoelectron line (blue) at a kinetic energy of 103.5 eV in agreement with the literature[39]. The width of about 4 eV does prevent us from distinguishing the spin-orbit splitting[39,40]. The photoelectron line is accompanied by shake-up satellites at around 91 and

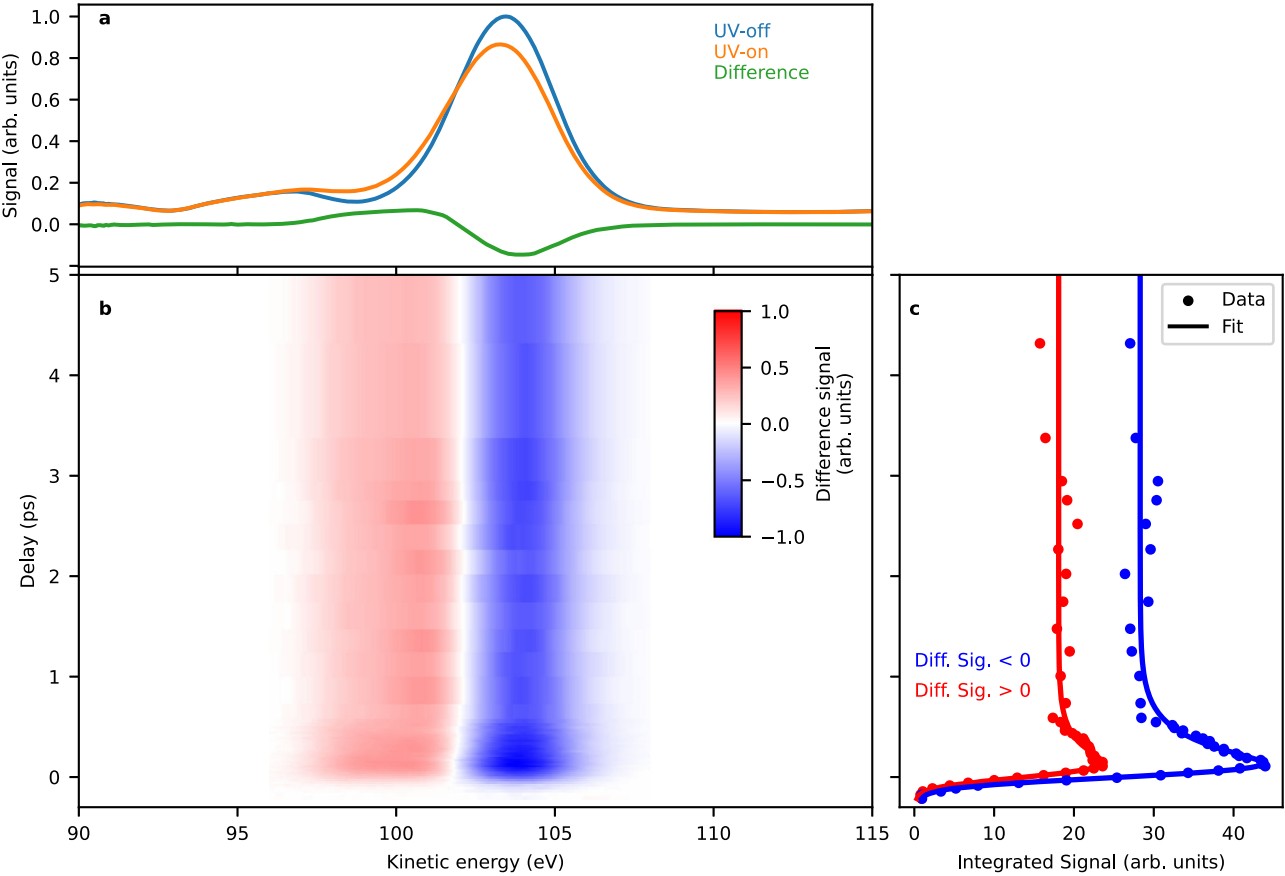

**Fig. 2 Experimental time-resolved XPS spectra of 2-thiouracil. a** UV-on (orange) and UV-off (blue) photoelectron spectra as well as the difference spectrum (green) between UV-on and UV-off at a delay of 200 fs. **b** False-colour plot of time-dependent difference XPS with red indicating UV-induced increase of the photoelectron spectrum and blue a UV-induced decrease. **c** Integrated signal of the positive (red) and negative (blue) parts of the difference spectra (dots) and fit to the data (solid line). Source data are provided as Source Data file.

96 eV[41]. Upon UV excitation ("UV-on", orange line), the $2p$-photoelectron line shifts towards lower kinetic energies. The difference spectrum ("UV-on" - "UV-off", green line at a delay of 200 fs) is equal to the difference between GS and ES spectra times the fraction, $f$, of excited molecules ($f \cdot$(ES-GS)). Part of the main photoelectron line is shifted into the region of the upper shake-up satellite, but the main part of the satellite line at 96 eV and the satellite at 91 eV remains unaffected. Figure 2b shows a time-dependent false-colour plot of the difference spectra. Temporal overlap has been determined by analysing the integrated absolute difference signal. The integrated signal under the positive/negative lobe in the difference spectrum is given in Fig. 2c.

The difference feature keeps its characteristic lineshape over the timescale of our measurement shown in Fig. 2b, indicating a persistent kinetic energy shift to smaller values over the whole range. The difference-amplitude changes significantly during the first picosecond. We use an exponential model function convoluted with a Gaussian time-uncertainty function of 190 (±10) fs FWHM (see Supplementary Discussion 1). We observe an exponential decay of 250 (±20) fs to 75% of the maximal signal for the negative part and 220 (±40) fs to 65% of the maximal signal for the positive part. The positive amplitude is always smaller than the negative amplitude. Systematic investigations of the difference spectra for various experimental settings exhibit the influence of so-called cyclotron resonances on the relative amplitudes in the MBES (see Supplementary Discussion 2). We therefore abstain from interpreting further the relative strength of the positive and negative features.

**Spectral oscillations at small delays.** Figure 3a shows a magnified part of the difference spectrum in Fig. 2b. To enhance the visibility of the spectral dynamics, we normalised each delay-slice on the area of the positive lobe. Despite the spectral width of about 4 eV, we identify oscillatory features in the positive part of the difference spectrum within the first ~600 fs. From zero delay to 150 fs, the spectrum shifts to lower kinetic energies and the peak of the spectrum widens. The shifts are most clearly visible in the spectral region from 99 to 101 eV. In the ground state, the shake-up peak at 96 eV has some spectral wing in this region. However, we do not observe a UV-induced change on the shake-up peak in its main part and lower energy wing. We thus assume that the main spectral effects in the 99–101 eV range are solely due to the UV-altered main photoelectron line.

After reaching minimal kinetic energies at 150 fs, the spectrum shifts towards higher kinetic energies by about 0.5 eV and the peak narrows reaching its extreme in the range between 200 and 300 fs. Subsequently, the spectrum shifts and widens again to reach its other extreme at 400 fs. For larger delays, the spectrum shifts again to higher kinetic energies. Further oscillations are not observed, however, for reasons of scarce experimental time, the delay steps are too coarse to follow additional oscillations (for more details see Supplementary Discussion 3). The negative lobe does not show systematic trends in this region and is therefore not shown in Fig. 3.

**Difference spectra simulations.** We will interpret the experimental results, casting them in the context of the rich literature

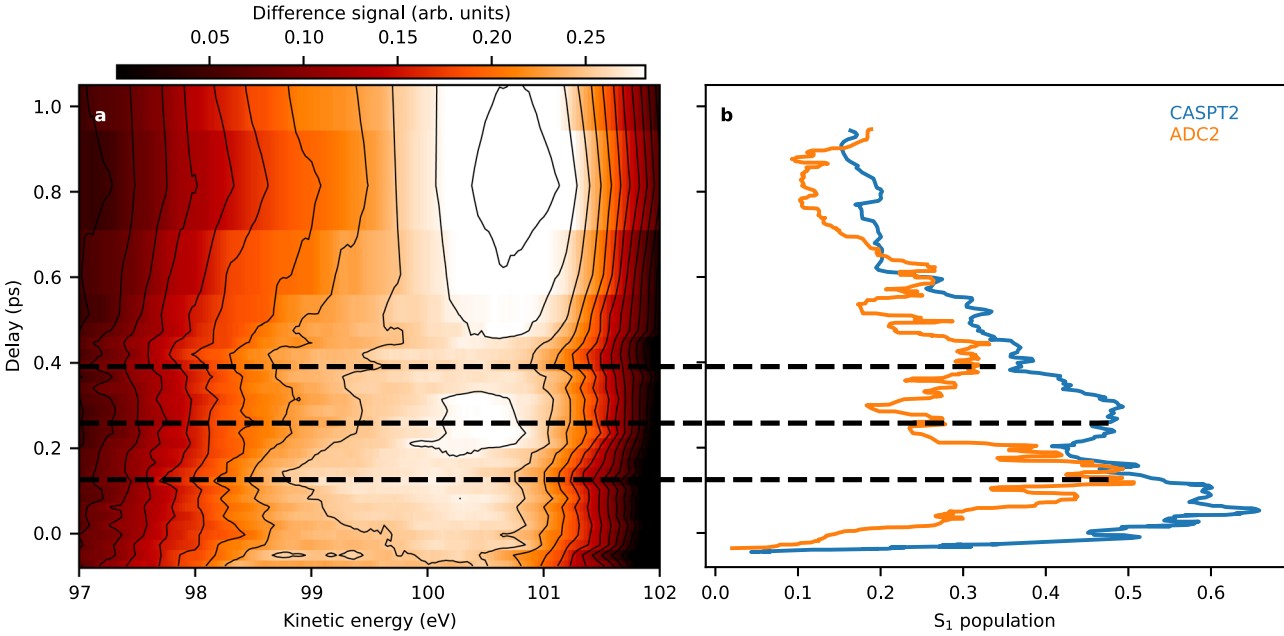

**Fig. 3 Experimental shifts and theoretical predictions on state population. a** False-colour contour plot of the positive lobe in Fig. 2b, normalised on the time-dependent area under the lobe. An oscillatory dynamic in the lineshape and position is visible for the first ~600 fs. At 150 fs and 400 fs delay, the spectrum is shifted to lower kinetic energy, while it is shifted to higher kinetic energies in between and afterwards. **b** Comparison of the oscillation dynamics with trajectory simulations. The population of the $S_1$ ($n\pi^*$) state, obtained from CASPT2 calculations of Ref. [35] (blue line) and ADC(2) calculations of Ref. [36] (orange line) are plotted. The dashed lines highlight the extrema of the oscillation observed in the experiment. The theoretical simulations do not include finite time-resolution and we shifted them by 50 fs to smaller delays to induce a transient rise of the signal around zero delay. The experimental 250 fs oscillation features have their counterparts in the simulated $S_1$ population, indicating the observation of a population exchange between the $S_1$ state and other electronic states. Source Data (for a) are provided as Source Data file.

on this molecule. Conclusions will also be drawn based on non-relativistic quantum chemical coupled-cluster calculations of the ground, valence-excited and core-ionised states of 2-tUra.

Previous calculations in the gas phase[34] or in the presence of water solvent molecules[42] identified global, non-planar minima, as well as nearly planar minima, which can be potentially visited by the photoinduced wave packet.

Along the same lines, we optimised the geometry of the lowest valence singlet ($S_0$, $S_1$, $S_2$) and triplet ($T_1$, $T_2$, $T_3$) states with and without the constraint of planarity. Since the $S_0$ minimum is nearly planar, the excited state (unstable) planar minima are likely to play a role in the short time dynamics and are marked with an asterisk in Figs. 4 and 5. Fully-optimised, stable non-planar minima could be found for the $S_1$, $S_2$ and $T_1$ states. The computational details are given in the Methods section and more extended in Supplementary Discussion 4. For all the geometries considered in this work the states $S_1$, $S_2$, $T_1$, $T_2$ and $T_3$ have $n\pi^*$, $\pi\pi^*$, $\pi\pi^*$, $n\pi^*$ and $\pi\pi^*$ character, respectively.

For each optimised geometry and each valence state we calculated the binding energy of the electrons in the three $2p$ core orbitals of the S atom. We estimated the ionisation cross sections as proportional to the norm of the associated Dyson orbitals and used this data to simulate pump-probe photoelectron spectra at different geometries. The stick spectrum (shown in the Supplementary Discussion 5) was convoluted with a Gaussian of a FWHM of 3.5 eV, to match the width of the experimental bands.

## Discussion

We first discuss the experimental data without reference to the XPS simulations. The difference spectra of Fig. 2, with their shift towards lower kinetic energies, indicate an increased $E_{bind}$ of the

UV excited states. The classical static XPS connects the $E_{bind}$ of a particular element on a particular site within a molecule to the total charge at the probed atom, which is related to the electro-negativity of the nearest neighbour atoms[5]. This connection is known as a 'chemical shift'. Accordingly, we anticipate that the CS can be generalised to a dynamic context as an ESCS in form of the difference of excited and ground state binding energy. The long-lasting shift of the kinetic energy would then indicate that the net effect of the photoexcitation is charge redistribution away from the sulphur atom (see Fig. 1). The electronic states inducing the strongest charge changes at the sulphur heteroatom are the $n\pi^*$ states. This is because the $n$ (lone-pair orbital) is strongly localised at the sulphur heteroatom (see Fig. 1b) and in the $n\pi^*$ states $n$ is singly occupied with respect to double occupation in the electronic ground state. Overall, this will lead to a strong ESCS to higher binding energies. The $^1n\pi^*$ state is generally considered a doorway state, leading from the UV excited $^1\pi\pi^*$ to the triplet states[37] and accordingly, we would expect an ESCS induced by this state.

The $\pi$ orbitals are less localised at the sulphur atom. Intuitively removal of an electron from a $\pi$ orbital would not induce as strong an ESCS as the $n$ removal, but it will still lead to some ESCS. These relative strengths of the ESCS can also be estimated by a simple charge analysis of the molecular wavefunctions, a method that can even be implemented with simple Hartree-Fock orbitals. We performed a Löwdin-population-analysis on the wavefunctions of the different electronic states at different geometries, which yields partial charges on the atoms of the molecule. In Fig. 4b, the calculated partial charge on the S atom is plotted against the calculated ESCS, which we will not use in the discussion yet as it is a much more complex entity to calculate. We clearly identify the strongest local positive charge on the sulphur atom with respect to the ground

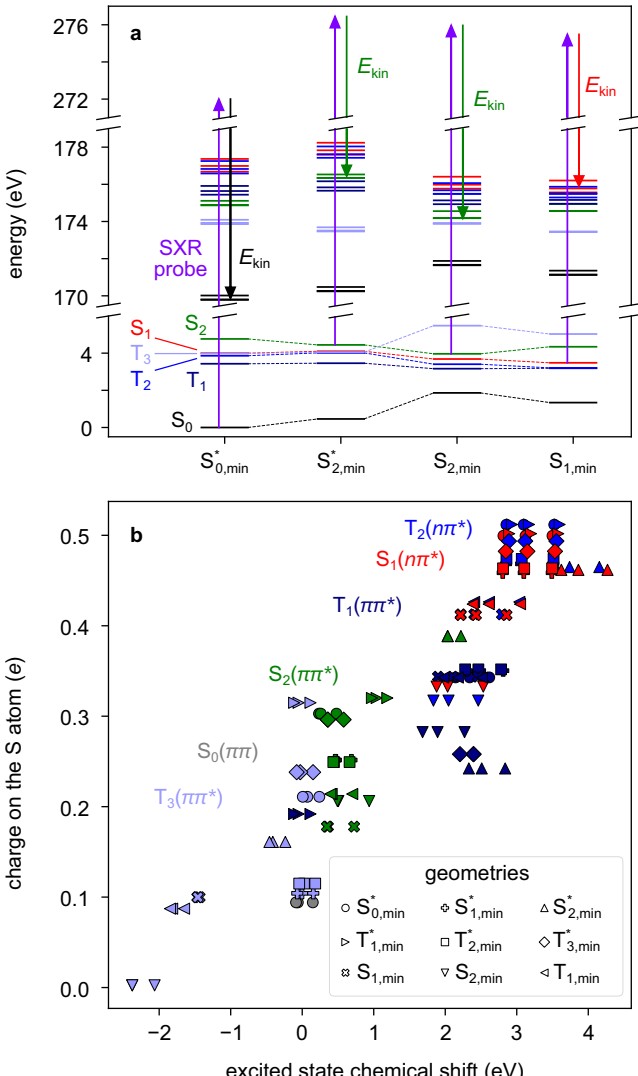

**Fig. 4 Soft x-ray photoelectron probing of the excited-state dynamics of 2-thiouracil in a multi-electron picture. a** Calculated electronic energies of the valence excited states, in the range 0–6 eV, and the core ionised states, in the range of 170–179 eV, for four geometries relevant in the short time dynamics. The arrows illustrate the $2p^{-1}$ ionisation process associated with the most intense transition. The core ionised states are grouped into sets of three states, which follow the colour coding of the valence states, meaning that their valence configuration is maintained with a $2p_x$, $2p_y$ or $2p_z$ core hole. Accordingly, the electron kinetic energy $E_{kin}$ refers to ionisations out of $S_0$, $S_1$ ($n\pi^*$) and $S_2$ ($\pi\pi^*$), depending on the geometry. **b** Partial charges on the S atom are plotted against the excited state chemical shift of the $2p$ electrons, calculated for the valence states at nine different geometries. For each geometry, the graph includes three markers for each excited state ($S_1$, $S_2$, $T_1$, $T_2$, $T_3$); in addition, three dots are included for the ionisation from $S_0$ at $S^*_{0,min}$, for a total of $9 \times 3 \times 5 + 3 = 138$ markers. Asterisks denote restriction in the calculation of the states to planar geometries. Source Data are provided as a Source Data file.

states in the $S_1$ ($^1n\pi^*$) and $T_2$ ($^3n\pi^*$) states, confirming the intuitive arguments given above. The $S_2$ ($^1\pi\pi^*$) state is characterised by a relatively low positive charge on the sulphur atom and the lowest $T_1$ ($^3\pi\pi^*$) state in its minimum geometry, generally considered as the long-lasting state[31,34], is lying in between the $n\pi^*$ and $S_2$ ($^1\pi\pi^*$) states. Thus, the permanent ESCS to lower kinetic energies (higher binding energies) is consistent with a relaxation cascade from the initially excited $S_2$

($^1\pi\pi^*$) over the $S_1$ ($^1n\pi^*$) and possibly also $T_2$ ($^3n\pi^*$) states into the lowest $T_1$ ($^3\pi\pi^*$) state.

Closer inspection of the short-time dynamics in Fig. 3a provides extraordinary experimental details on the molecular relaxation dynamics. In the spectral shift dynamics, the strongest ESCS to lower kinetic energies (higher binding energies) is found at 150 and 400 fs, interrupted with an interval and followed by delays of smaller ESCS. Intuition based on the lone-pair orbital localisation, as well as the simple local charge analysis at the sulphur atom mentioned above, indicate that this spectral shift reflects changes in the electronic state of the molecule with maximal $n\pi^*$ contributions at 150 and 400 fs and consequently minimal $n\pi^*$ contributions in between and after. We compare theoretical predictions of the $S_1$ ($^1n\pi^*$) state population dynamics from the trajectory surface-hopping calculations of Mai et al.[35,36]. in Fig. 3b. The $S_1$ populations calculated using CASPT2 and ADC(2) potentials show indeed dynamics that fit well to the experimentally observed shifts. The oscillation period depends on the theoretical approach, and the timing of the second $S_1$ population maximum fits the experimental data better in the case of the ADC(2) approach. The simulations of Mai et al. predict a population transfer among the $S_1$, $S_2$ and triplet states in a coherent fashion and the details of states participating in the population dynamics depends on the applied electronic structure method. However, in both cases, the $S_1$ state, having the highest IP of all states, carries the largest oscillation. This comparison strongly supports the experimental arguments for observing population dynamics entering and leaving the $S_1$ state in a coherently modulated fashion.

Similar coherent modulations have been observed in the transient spectra for 4-tUra and 2-tUra and in liquid phase[42,43]. In case of 4-tUra, much faster, sub-100 fs, coherent modulations have been observed in fs transient absorption spectra and attributed to particular vibrational modes in the electronically excited state of 4-tUra[43]. For the case of 2-tUra, the same experimental methods exhibit oscillations with a period very similar to ours[42], which are however not interpreted. We also checked the assignment to a purely vibrational coherence by performing a normal mode analysis at the calculated excited state minima. Only at the nonplanar minimum of the $S_2$ ($\pi\pi^*$) state we found a mode with a frequency of 131 cm$^{-1}$, thus compatible with a 250 fs modulation. However, this would mean that the molecular population should be dominated by the $S_2$ state for the 600 fs of our modulated time-interval, which is in contrast to all current literature suggestions. Also, for short times, we suggest that the molecule remains mostly planar on the $S_2$ state, as will be shown below by comparison to calculated difference spectra. Remarkably, valence photoelectron spectra of 2-tUra in the gas phase do not show these modulations[31], demonstrating that the ESCS in XPS is able to pick up molecular dynamics beyond reach for valence electron photoemission.

On top of the coherently modulated signal, we find an amplitude decay with a time-constant of 220–250 fs, being equal on the positive and negative lobe of the difference spectra within the error bars. Similar short decays have been observed in liquid and gas phase studies of 2-tUra. In valence photoelectron spectroscopy, a time constant of around 300 fs is observed for the excitation wavelength corresponding to the one used here[31]. Based on the calculated valence IPs of the different states[44], the decay is attributed to relaxation from the $S_1$($^1n\pi^*$) to the triplet state manifold. Liquid-phase transient absorption spectra of 2-tUra in the same work show similar time constants and the same interpretation is applied[31]. This is supported by a recent joint experimental-theoretical investigation that includes calculated transient absorption windows at some of the molecule's crucial excited state positions[42]. The observed vanishing contrast in our TR-XPS might need to be interpreted in a different way

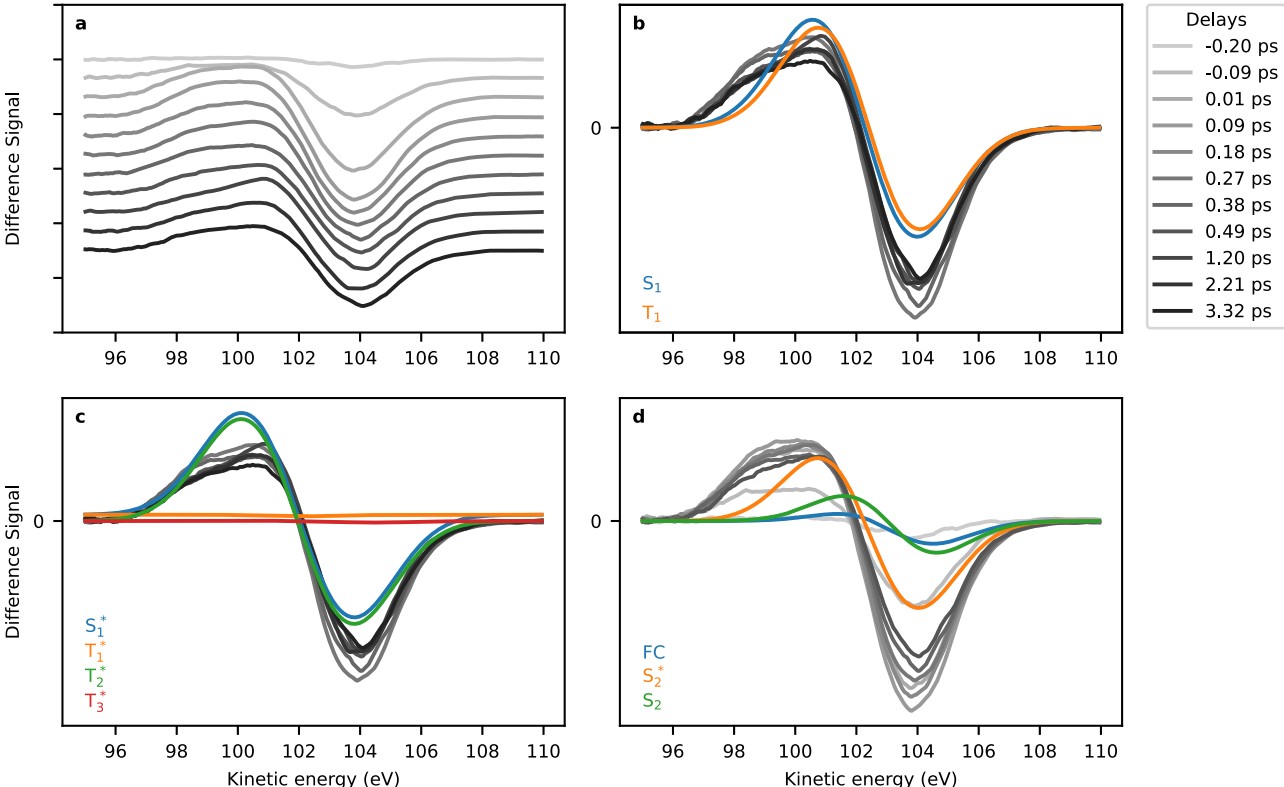

**Fig. 5 Comparison between experimental and theoretical difference spectra. a** Ridgeline plot of experimental difference spectra for different pump-probe delays between −0.2 and 3.32 ps. **b–d** Comparison of experiment (greyscale lines) with theoretical (coloured lines) difference spectra computed at the coupled-cluster level of theory. All calculated difference spectra are shifted lower in kinetic energy by 1.3 eV. **b** and **c** Spectra with delays in the range 0.27–3.32 ps, (**d**) spectra with delays −0.2–0.49 ps. The theoretical spectra use the energies at the minima ($S_2$ is the nonplanar, $S_2^*$ is the restricted geometry) and the Franck–Condon geometry. Asterisks denote restriction in the calculation of the states to planar geometries. Source Data are provided as a Source Data file.

than the $S_1(^1n\pi^*)$—triplet state relaxation, as we would expect to still observe a ESCS leading to different features in the triplet manifold. The state diminishing the modulation contrast of the difference spectra would have to show a small ESCS with respect to the ground state, and thus be electronically similar—or identical—to the ground state. We thus suggest an ultrafast molecular decay of part of the population into the electronic ground state and check it further by comparison to ESCS simulations below.

We now discuss the calculated ESCS and its connection to the charge on the sulphur atom in the photoexcited states. We observe the remarkable linear relation between binding energy and charge, closely resembling the potential model introduced in static XPS[6]. While many effects such as final-state charge relaxation and core-hole screening[13,45] shape the calculated binding energy, the linear trend prevails. This provides a generalised concept for deducing local charge changes in electronically excited states from the ESCS. Further well-established corrections known from static XPS to this plot make the $E_{bind}$ - charge connection even more obvious (Supplementary Discussion 6).

In addition, we can clearly distinguish drivers behind the charge and ESCS shifts. The colours in Fig. 4 indicate the electronic state while the different symbols indicate geometries, corresponding to minima or saddle points on different potential energy surfaces. We clearly observe clustering according to electronic state, although very widely differing geometries have been used. Thus, the main factor for local charge and therefore binding energy is the electronic state of the molecule. This in turn gives core-level photoelectron spectroscopy high electronic state sensitivity. The exceptions of one $S_2$ and $T_1$ geometry are explained

in the Supplementary Discussion 7. The data for the $S_1$ and $T_2$ states ($n\pi^*$) are very well clustered in the upper right corner of Fig. 4b, with the highest ESCS. As explained above, these states have $n\pi^*$ character and possess a high positive charge on the sulphur atom, which is also illustrated in Fig. 1.

In Fig. 5, we assess details of the TR-XPS spectra by comparison with the calculated spectra, extending our discussion of the purely experimental features. The onset of the experimental difference feature around time-zero is shown in more detail in the ridgeline plot of Fig. 5a. The difference signal gets stronger with delay indicating an increasing $f$ over the time-resolution of 191 fs. We initially identify the development of the negative feature sitting stable at 104 eV and a comparatively broader positive feature at lower kinetic energies. The relative position of the positive and negative bands is very well captured by the simulation. Since similar Dyson norms were predicted for the different initial valence excited states, the calculated spectra essentially integrate to zero. According to model and intuition, the positive feature should be attributed to electronic states with higher positive charge on the sulphur atom.

We continue discussing the picosecond difference spectra, comparing them to calculated difference spectra in Fig. 5b and c. Among the states with highest local charge and thus highest $E_{bind}$ are the $S_1(n\pi^*)$, $T_1$ ($\pi\pi^*$) and $T_2$ ($n\pi^*$) states. The unstable, restricted planar minima/saddle points of $S_1$ and $T_2$ (Fig. 5c) could be responsible for parts of the difference spectra for a limited time until relaxation into a non-planar geometry. Among these non-planar geometries, both the $S_1$ and $T_1$ show agreement with the experimental difference spectra (Fig. 5b). The missing low-energy wing from 97 to 98.5 eV in the theoretical spectra can

likely be attributed to the fact that we neglect wave-packet and incoherent thermal distribution effects resulting in extended geometry coverage. The shake-up phenomena are also not modelled. However, we do not observe a change of the shake-up peaks and we would expect that the observation of UV-induced effects on the weaker satellite lines requires better signal-to-noise ratio. The theoretical simulations strengthen the arguments given in the context of Fig. 3, showing that a strong ESCS due to the $S_1$ coherent dynamics is followed by a state with less ESCS. We note that simulations/experiments on the pyrimidine nucleobases call for the $S_1$ $^1n\pi^*$ state to be occupied first after $S_2$ relaxation[46]. Only thereafter will the triplet states gain population. Based on the experimental features, we can exclude the planar geometries of the $T_1$ and $T_3$ states to play a major role in the relaxation process. A comparison to the ESCS in Fig. 4b implies that the charge at the sulphur atom has reached about half a positive value due to the molecular relaxation.

We now return to the initial 220–250 fs decay of the difference feature, which can be an indication of relaxation of the charge back to the ground state distribution, into states with negligible photoionisation cross section, or into states with an IP equal to the electronic ground state before photoexcitation. To obtain more insight, we compare the experimental XPS to simulations in Fig. 5b–d. Figure 5d shows the $S_2$ ($\pi\pi^*$) state difference spectra in three different molecular geometries. Figure 5c shows spectra with 'unstable' planar geometries. The most important is the $S_1$ ($n\pi^*$) state, others are expected to play a minor role in the dynamics for short times[35,36]. Figure 5b shows difference spectra for the $S_1$ and $T_1$ states at their respective potential energy minimum geometry. The unstable $T_1^*$ and $T_3^*$ spectra are indeed flat on our scale and could in principle explain the observed decay in difference amplitude. However, these geometries cannot be stable, and cannot explain the long-lasting reduction in amplitude with the 220–250 fs exponential decay. In addition, the remaining calculated non-flat difference spectra do not exhibit a large enough difference. We therefore attribute the observed decay to an ultrafast relaxation into the electronic ground state with net zero charge change with a 220–250 fs time-constant.

Ground-state (GS) relaxation has been discussed before in the solution as well as in the gas phase[31,37,47]. In earlier solution-phase work, a remark on an incomplete triplet yield could be interpreted in favour of an ultrafast ground state decay channel[37]. More recent work in the gas phase has attributed time-constants from 50 to 200 ps to ground state relaxation[31]. Our ground state channel is close to the 300 fs decay observed in Ref. [31]. which was attributed to intersystem crossing previously.

We now compare early difference spectra to the calculated spectra of the directly photoexcited $S_2$ ($\pi\pi^*$) state in different geometries in Fig. 5d. The best representation is given by the $S_2^*$ planar geometry. The Franck-Condon spectrum should be included in the difference spectra, however, the short lifetime of this point as compared to our time-resolution makes it a minor contributor. We can, however, exclude a relaxation with major contributions from the $S_2$ out-of-plane minimum (green line), as this would mean a shift in the zero-crossing feature by about 1 eV to higher kinetic energies, which is not observed in the experiment. This observation agrees with the predictions of trajectory calculations when using a CASPT2 approach to electronic structure[35] but is in contrast to trajectory calculations with an ADC(2) electronic structure approach[36]. While in the first reference the $S_2$ lifetime is below 100 fs and thus too short for the molecule to effectively reach out of plane geometries, the latter predicts an $S_2$ lifetime of 250 fs which allows for out of plane geometries.

In a recent theoretical-experimental solution phase study, two $^1\pi\pi^*$ states with a slight energy gap are assumed to be populated and while the upper one is predicted to relax via out-of-plane geometries, the lower one is suggested to relax via planar geometries[42]. We find that our calculated geometries are very similar to the ones from Ref. [42]. (see the Supplementary Discussion 4 for a full comparison) and thus our analysis above would clearly advocate for initial planar geometries in the relaxation path. However, as our study is performed on isolated molecules instead of in solution phase it is not clear if we have any appreciable admixture of the higher lying $^1\pi\pi^*$ state.

We have introduced the concept of ESCS in TR-XPS and shown that this powerful concept can be applied to deduce charge distribution changes in excited molecules. We observe rich dynamics on a sub-ps timescale. Based on intuitive arguments, we can assign the spectral features to coherent population exchange of the $n\pi^*$ state, inducing a strong ESCS, with other electronic states of less ESCS. In addition, we identify an ultrafast ground state relaxation path based on decaying amplitude in the differential signal. The calculated ESCS as a function of electronic state and geometry help in interpreting the geometric changes of the molecule after UV excitation in terms of a planar relaxation path on the photoexcited $^1\pi\pi^*$ state. The connection between charge change at the probe site and the exactly calculated ESCS can be well approximated by a potential model, as in ground state XPS. This will provide a methodological basis for an intuitive understanding of charge dynamics in photoexcited isolated molecules on the femto- and attosecond timescale but also for photocatalytic systems. In a next step, one can address more than one site of the molecule using TR-XPS and map the charge flow induced by photoexcitation over the whole molecule.

## Methods

**Time-resolved UV pump soft x-ray probe (photo-) electron spectroscopy**. The experiment was performed at the FL24 beamline of the FLASH2 facility at DESY using the newly built URSA-PQ apparatus. A detailed description of the apparatus and the experiment can be found elsewhere[48,49]. In short, the apparatus includes a magnetic bottle time-of-flight electron spectrometer (MBES), a capillary oven to evaporate the 2-thiouracil samples at 150 °C and a paddle with beam diagnostics on top of the oven. UV pump pulses of 269 nm centre wavelength, 80 fs duration and an energy around 1 µJ were focused to a 50 µm focus to pre-excite the molecules into the $\pi\pi^*$ state. Power scans on the time-dependent spectral features were performed to assure that the signal is not over-pumped by the UV pulses (Supplementary Discussion 8).

Tunable soft x-ray pump pulses were produced in form of SASE (self-amplified spontaneous emission) radiation. Every second x-ray pulse was delivered without UV excitation for obtaining a reference on the non-excited molecule. The mean x-ray photon energy was set to 272 eV with a bandwidth of 1–2% (including jitter). The x-ray probe was linearly polarised parallel to the axis of the magnetic bottle spectrometer and the UV polarisation. The focal size of the x-ray beam was slightly larger than the UV spot size. Systematic power scans were performed to exclude nonlinear effects in the x-ray induced electron spectra (see Supplementary Discussion 8). To increase energy resolution of the spectrometer, the speed of the ejected electrons was reduced by an −80 V retardation voltage on an electrostatic lens in front of the 1.7 m long flight tube which was kept at a constant potential. The energy resolving power of the MBES ($E/\Delta E$) has been determined with Kr MNN Auger lines to be 40 at 0 V retardation. Based on the sulphur 2p-photoelectron line, we estimate the resolution to be better than 30 with respect to the total kinetic energy. The time-dependent spectra were measured for a series of delays. In each scan, the delays were set randomly to avoid systematic effects. We measure the difference spectra of UV excited to non-excited shots. The data evaluation is described in Supplementary Discussion 9.

**Theoretical calculations**. The geometry optimisation of ground state 2-thiouracil was performed using coupled-cluster theory with singles and doubles (CCSD) with the 6–311++G** basis set[50,51]. Valence excited state optimisations and energy calculations at specific geometries were performed using the equation-of-motion formalism (EOM-CCSD) with the same basis set. All calculations were performed using the package Q-Chem 4.4[52]. At all computed geometries the valence excited states' wavefunctions are dominated by a singly-excited configuration. For the states $S_1$, $T_2$ ($n\pi^*$) and $S_2$, $T_1$ ($\pi\pi^*$) the involved orbitals at the Franck-Condon point are shown in Fig. 1. The structural parameters of the optimised planar and non-planar geometries on the different electronic states are reported in the Supplementary Discussion 4.

The binding energy of the electrons in the 2p orbitals of the sulphur atom were calculated using the equation-of-motion coupled-cluster method for IPs (EOM-IP-CCSD)[53] with the 6–311 + +G** basis set for the S atom and the 6–31 + +G basis for all other atoms. Photoelectron intensities were approximated as the geometric mean of the norms of the left and right Dyson orbitals associated with the ionisation process[10]. The target core-excited states of the cation were identified as the eigenstates which have the largest overlap with initial guess states, obtained by applying the annihilation operator of the three 2p electrons on reference CCSD wavefunctions, according to the procedure implemented in Q-Chem. To this end, reference CCSD wavefunctions for the different electronic states of the neutral molecule were calculated starting from unrestricted Hartree-Fock wavefunctions, which were optimised using the maximum overlap method, in order to mimic the (singly excited) orbital occupancy of the excited states. A similar strategy has been recently validated by Coriani et al. to model pump-probe x-ray absorption spectra of nucleobases at the coupled-cluster level[54]. Spin-orbit coupling, leading to a splitting of the core-ionised states of the order of 1 eV (not resolved due to spectral broadening) is not included in the calculations.

## Data availability
Source data are provided with this paper. The FEL raw data, several TB in size, that support the findings of this study are available from the corresponding authors upon request. The processed photoelectron spectra are provided in the Source Data File. The results of the theoretical calculations i.e. optimised geometries, ionisation potentials, state energies and partial charges, are provided in the Source Data File. Source data are provided with this paper.

## Code availability
The codes used to generate the computational results of this study are highly adapted to the hdf5 output of the FLASH free-electron laser. They are available from the corresponding authors upon request.

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

## Acknowledgements

We thank the Volkswagen foundation for funding via a Lichtenberg Professorship. We thank the BMBF for funding the URSA-PQ apparatus and for funding J.M. via Verbundforschungsprojekt 05K16IP1. We acknowledge DFG funding via Grants GU 1478/1-1 (M.G.) and SA 547/17-1 (P.S.). T.J.A.W. was supported by the US Department of Energy, Office of Science, Basic Energy Sciences, Chemical Sciences, Geosciences, and Biosciences Division. R.F. thanks the Swedish Research Council (VR) and the Knut and Alice Wallenberg Foundation, Sweden, for financial support. We acknowledge DESY (Hamburg, Germany), a member of the Helmholtz Association HGF, for the provision of experimental facilities. Part of this research was carried out at FLASH2. F.C. acknowledges support from the European Research Council under the ERC-2014-StG STARLIGHT (Grant Agreement No. 637756).

## Author contributions

J.M. and M.G. did the conceptual design for the experimental apparatus. J.M.designed and built the apparatus. The magnetic bottle spectrometer was designed by J.M. with help of R.F. and R.J.S. and M.G. J.M., D.M., F.L. and S.D. integrated the apparatus into the FLASH beamline. F.L. designed and executed the data interfacing between the experiment and the FLASH data infrastructure. D.M., F.L., J.M., S.A., F.C., S.D., R.F., M.N., B.M., M.K., T.M., M.S.R., R.J.S., A.T., M.W., T.J.A.W. and M.G. performed the experiment. D.M. and F.L. analysed the data. D.P. performed the calculations and discussed the theoretical methods with C.E. and P.S. D.M., F.L., D.P. and M.G. discussed the data and theory and prepared the paper. All authors contributed to the final version of the paper by discussions and/or edits. M.G. wrote the beamtime proposal for this experiment and serves as spokesperson for the collaboration.

## Funding

## Competing interests

The authors declare no competing interests.
