## [Peer Review File · Nature Communications]

Parts of this peer review file have also been redacted as indicated to maintain the confidentiality of other journals.

Reviewer #2 (Remarks to the Author):

Mayer, *et al.* describes an advance time-resolved X-ray photoelectron spectroscopy (TR-XPS) experiment on the gas phase molecule thiouracil. The experiments are well described and the analysis detailed. As a first fully time resolved measurement TR-XPS measurement on the lighter atoms more associated with organic chemistry. The experiments are therefore very interesting and could be impactful. I have 2 main questions about the analysis and interpretation that require clarification before I could recommend acceptance.

1. The analysis and interpretation require some clarification or extra detail (either in the manuscript or the SI) to be convincing. A large amount of the analysis is related to the shifts in the centre of one fitted Gaussian to a distribution. While I understand that this is empirical and a way to quantify changes in the spectrum there is currently no evidence presented on how well this fits the spectral shape? if both gaussians have variable width and centre of if this is fixed on one and not the other? Adding some details of the fit parameter, why the parameters of only one gaussian are presented, the quality of the fits or the residuals in the SI may help convince me that the shifts are significant. It is also not explained how the error bars on the centre positions are obtained or if these are at what level these are given.

The lack of detail on these aspects makes me wonder if shift on the order of 0.2 eV on a peak of 4 eV width is reliable and significant.

2. The discussion on the ESCS classifies states in terms of electronic character (e.g. $n\pi^*$) which are given their equivalent adiabatic labels (e.g. S1) presumably related to equilibrium geometry of the electronic ground state. The discussion of the simulations and the dynamics then discuss using the adiabatic labels and the populations are discussed accordingly. As the electronic states cross and change energy with geometry presumably the energy ordering changes allowing for population transfer and changes in electronic character as states cross.

Can we really read the S1 population as meaning $n\pi^*$ population for all geometries as suggested in the manuscript?

If S1 always relates to the $n\pi^*$ state and all other labels remain the same: As it is this electronic character that defines the ESCS can I suggest that these labels are used throughout the manuscript, this may help with the explanation of figure 4 and provide a clearer argument for the electronic character being responsible for the excited state shift.

If S1 is not always the $n\pi^*$ state: This would make the argument around the similarity in oscillation period seen with the S1 population being coincidental or due to a different origin than that explained in the earlier description. If the S1 label is not equivalent to $n\pi^*$ then the authors could consider plotting against $n\pi^*$ character instead if this is possible?

If these points can be clarified then the manuscript could be suitable for publication in nature communications.

Reviewer #3 (Remarks to the Author):

I have already reviewed this paper for [redacted] and I see that the authors have made a significant effort at improving the presentation and giving a physical picture of the on-going processes prior to backing them with theory. The paper reads nicely and the flow of reasonings is sound and logical.

I still have a comment though: In figure 2, I can see a shift but I am surprised that the integrated intensity of the positive and negative parts of the green trace do not sum up to zero. The authors attribute this to the influence of cyclotron resonances, but the inequivalent intensities of the positive and negative parts seem to be captured by their calculations in figure 5. Could they comment on this?

A point of detail: ref. 26 does not concern X-ray absorption spectroscopy.

Potsdam, 23/9/2021

Responses:

Our response in red, changes in the manuscript are marked yellow, citations from the manuscript in italics.

Referee 2:

Mayer, et al. describes an advance time-resolved X-ray photoelectron spectroscopy (TR-XPS) experiment on the gas phase molecule thiouracil. The experiments are well described and the analysis detailed. As a first fully time resolved measurement TR-XPS measurement on on the lighter atoms more associated with organic chemistry. The experiments are therefore very interesting and could be impactful. I have 2 main questions about the analysis and interpretation that require clarification before I could recommend acceptance.

We thank the referee for appreciating the work on lighter atoms as well as stating that this could have impact.

1. The analysis and interpretation require some clarification or extra detail (either in the manuscript or the SI) to be convincing. A large amount of the analysis is related to the shifts in the centre of one fitted Gaussian to a distribution. While I understand that this is empirical and a way to quantify changes in the spectrum there is currently no evidence presented on how well this fits the spectral shape? if both gaussians have variable width and centre of if this is fixed on one and not the other? Adding some details of the fit parameter, why the parameters of only one gaussian are presented, the quality of the fits or the residuals in the SI may help convince me that the shifts are significant. It is also not explained how the error bars on the centre positions are obtained or if these are at what level these are given.

We thank the referee for this important comment.

We agree with the referee, that we need to give more information about the Gaussian fit. We have therefore expanded Section 1 of the Supporting Information containing all necessary information to assess the quality of the fit. We added parameters of both Gaussians fitted. Regarding the question, how the fit would behave if one of the two Gaussians would be fixed, we can make a definite statement without trying a new fit. The new Figure S1 shows the Gaussian center, and width for the free fit shown in the paper. We clearly identify that the fit of the negative lobe does not undergo major spectral shifts or shifts in the width compared to the positive lobe. Therefore, the fit really picks up the spectral changes only observed in the positive lobe in the correct way.

Regarding the error bars given in Fig. 3, we have used the standard error of the Gaussian peak as given by the fit.

Regarding the question to assess a measure of the fits quality, we have investigated the residual:

Here we show the residual of the fit. As stated in the paper and by the referee, the fit is empirical and we can see that the Gaussian fit of the positive lobe does not have the correct analytical shape, as at the low and high energy side signal is not described completely by a single Gaussian, but this is also not the purpose of the fit. Instead it is the simplest model to pick up the spectral shift. We plot the development of the residual as a function of time in the following graph:

At the critical point of the coherent oscillation of the spectral mean, no systematic shifts are visible that for instance show the oscillation identified in the paper, however out of phase, so that the two effects would cancel. We are therefore confident in the unambiguity of the sift shown in the paper.

To make this transparent to the reader, we added the following section to the SI of the paper:

To quantify the shifts in the positive and negative lobe of the delay-dependent difference spectrum, we fitted a sum of two Gaussians to the individual spectra. For the fit neither the central position nor the width of the Gaussians have been fixed. The resulting central positions (top) as well as the Gaussian width (bottom) for both lobes are shown in figure S1 together with their fitting errors. The negative lobe in blue barely shows any movement in both parameters while scanning the delays. The positive lobe in red, however, shows a distinct oscillation for delays between 0 and 500fs in both position and width.

The Gaussian model has its shortcomings as mostly the positive line is not a perfect Gaussian. However, the residuals of the fit are small compared to the overall signal and we could not identify any systematic trends that would explain the observed oscillation based on an inadequate fitting model. The Gaussian functions manage to follow the center of mass of the two lobes similar to what can be observed in figure 3 (a) or 5 (a) in the main text.

Figure S2: Central positions and width of the two Gaussians fitted to the delay-dependent difference spectra. The given error bars are the standard errors resulting from the fit.

The lack of detail on these aspects makes me wonder if shift on the order of 0.2 eV on a peak of 4 eV width is reliable and significant.

The referee states correctly that this is an empirical way to describe the results. The main features we describe by the fit are very well visible without fitting in the false color plot of figure 3a. The color as well as the contour lines clearly show an oscillatory motion of the photoelectron band. The significance of the shifts does not rely on the Gaussian fitting but is visible from the raw data in an unambiguous manner.

We changed the wording in the paper accordingly to pronounce this point:

'...The negative lobe does not show systematic trends in this region and is therefore not shown in Fig. 3.

The shifts are visible already in the raw data in Fig. 3 as described above, nevertheless we employ a fit to reduce the complexity of the raw data. The simplest fitting model able to pick up the spectral fluctuations is a double Gaussian fit of the whole difference spectrum.....'

It is true that a 0.2 eV shift on a broad line is a small effect, however the good statistics of the experiment makes this attribution possible, as the small standard errors and the modulation of only the positive Gaussian fit for a completely free two-Gaussian fit demonstrates. We hope that the additional information in the SI is convincing regarding the fit parameters.

2. The discussion on the ESCS classifies states in terms of electronic character (e.g. $n\pi^*$) which are given their equivalent adiabatic labels (e.g. S1) presumably related to equilibrium geometry of the electronic ground state. The discussion of the simulations and the dynamics then discuss using the adiabatic labels and the populations are discussed accordingly. As the electronic states cross and change energy with geometry presumably the energy ordering changes allowing for population transfer and changes in electronic character as states cross. Can we really read the S1 population as meaning $n\pi^*$ population for all geometries as suggested in the manuscript?

If S1 always relates to the $n\pi^*$ state and all other labels remain the same: As it is this electronic character that defines the ESCS can I suggest that these labels are used throughout the manuscript, this may help with the explanation of figure 4 and provide a clearer argument for the electronic character being responsible for the excited state shift.

If S1 is not always the $n\pi^*$ state: This would make the argument around the similarity in oscillation period seen with the S1 population being coincidental or due to a different origin than that explained in the earlier description. If the S1 label is not equivalent to $n\pi^*$ then the authors could consider plotting against $n\pi^*$ character instead if this is possible?

We would like to thank the referee for pointing this out. We have not been completely consistent in our nomenclature, as here the ' S_x ' ' T_x ' nomenclature and the electronic orbital nomenclature do not change for the different geometries explored. We now state this clearly in the paper:

'...The computational details are given in the Methods section and more extended in Supplementary Discussion 3. For all the geometries considered in this work the states S_1 , S_2 , T_1 , T_2 and T_3 have $n\pi^$, $\pi\pi^*$, $\pi\pi^*$, $n\pi^*$ and $\pi\pi^*$ character, respectively.'*

We followed the suggestion of the referee and now added the orbital labels to the ' S_x ' ' T_x ' nomenclature at every point in the paper also in Figure 4b, as intended by the referee.

If these points can be clarified then the manuscript could be suitable for publication in Nature Communications.

We thank the referee for all the work on the paper, the great suggestions and we believe that this really improved the paper!

Referee 3:

I have already reviewed this paper for [redacted] and I see that the authors have made a significant effort at improving the presentation and giving a physical picture of the on-going processes prior to backing them with theory. The paper reads nicely and the flow of reasonings is sound and logical.

We thank the referee for this positive assessment of our paper.

I still have a comment though: In figure 2, I can see a shift but I am surprised that the integrated intensity of the positive and negative parts of the green trace do not sum up to zero. The

authors attribute this to the influence of cyclotron resonances, but the inequivalent intensities of the positive and negative parts seem to be captured by their calculations in figure 5. Could they comment on this?

The calculations show equal positive and negative contributions. Maybe the last version of Figure 5b led the referee to this conclusion, as we accidentally had the zero line offset (the 0 was written at the wrong tick on the axis. In all other parts of the figure the 0 was at the right position). This is now corrected. We are very sorry about this error.

We also added a sentence about the equal positive and negative parts in theory in the main text:

'In Fig. 5, we assess details of the TR-XPS spectra by comparison with the calculated spectra, extending our discussion of the purely experimental features. The onset of the experimental difference feature around time-zero is shown in more detail in the waterfall plot of Fig. 5a. The difference signal gets stronger with delay indicating an increasing f over the time-resolution of 191 fs. We initially identify the development of the negative feature sitting stable at 104 eV and a comparatively broader positive feature at lower kinetic energies. The relative position of the positive and negative bands is very well captured by the simulation. Since similar Dyson norms were predicted for the different initial valence excited states, the calculated spectra essentially integrate to zero. According to model and intuition, the positive feature should be attributed to electronic states with higher positive charge on the sulphur atom.'

A point of detail: ref. 26 does not concern X-ray absorption spectroscopy.

We thank the referee for this comment. We changed the sentence with respect of this reference: 'Hard x-ray absorption and emission spectroscopy is highly sensitive to charge and spin states, however, only on metal atoms within molecules^{25,26}.'

We are also grateful for all comments by referee 3 and believe that they made the paper much more accessible.

Reviewer #2 (Remarks to the Author):

The authors have answered all of my questions however, I remain unconvinced that the reported shifts are significant. The plot provided in the rebuttal are instructive. The residual plot at a given delay highlights the limitations of the fit of which the authors provide an honest appraisal, and while at first glance the time dependent residuals look convincing, the majority of the plot is over regions where an oscillation is not observed.

Looking closely at the first 500 fs or so, where the oscillation in the manuscript are reported, there appear to be similar shifts in the residual. Taking the new figure in the rebuttal (copied below) and focusing on the residual peak around 100 eV (close to the centres reported in the main manuscript), then there appears to be an initial blue shift to higher eke which opposes the shift presented in the fit in the paper. The scale of the figure then makes comparison difficult but it appears subsequent changes are also present.

I therefore remain unconvinced that the simple fitting procedure used to extract the shifts is doing a reliable job. There are clear systematic differences in the peak shape used in the fit compared to what is observed such that I do not believe the gaussian centre and width provides an accurate description of the changes occurring. As this part of the analysis is the cornerstone on which much of the subsequent discussion and interpretation relies, I currently cannot recommend publication.

Reviewer #3 (Remarks to the Author):

The authors have addressed my last questions. The paper is now fit for publication.

Response to the referees

We want to thank the referees very much for the detailed referee responses. We appreciate the hard work that went into that, and we are happy that referee 1 and referee 3 are suggesting that this paper is of high value for the community and appropriate for publication in Nature Communications. We are also happy that we could convince referee 2 about part of their questions.

We would like to use the opportunity to convince the referee also about the other part of the questions raised.

Essentially, the remaining doubts of referee 2 touch the applicability of the Gaussian model. In the following, we extend our discussion of the Gaussian model, to show that one 'could' indeed use it. However, as already stated in the main text of the manuscript, one does not need a Gaussian fitting model to arrive at our conclusions, and we now completely cancel any mention of the Gaussian model from our manuscript, without changing anything but the caption of fig. 3 and removing the mention of the Gaussian model below figure 3.

In the following we first discuss the answers to the questions of referee 2 and then show changes in the main text and supporting information part of the manuscript.

The referee states that:

'Looking closely at the first 500 fs or so, where the oscillation in the manuscript are reported, there appear to be similar shifts in the residual. Taking the new figure in the rebuttal (copied below) and focusing on the residual peak around 100 eV (close to the centres reported in the main manuscript), then there appears to be an initial blue shift to higher eke which opposes the shift presented in the fit in the paper.'

We are very sorry that the referee had to use the large residual map we delivered last time. It spanned 2ps in time and over 10 eV in kinetic energy range. We now deliver a residual map with the appropriate range. The comparison with Fig. 3 in the main text shows are no 'similar shifts' observable in the residual that change the experimental feature, the 250 fs oscillation along the time axis.

We wrote about this in the paper: '...After reaching minimal kinetic energies at 150 fs, the spectrum shifts towards higher kinetic energies by about 0.5 eV and the peak narrows reaching its extreme in the range between 200 to 300 fs. Subsequently, the spectrum shifts and widens again to reach its other extreme at 400 fs. ...'

The residual in part b) shows stripes that are one order of magnitude lower than the signal itself. Please note that the colorbar legends in a) and b) are not directly comparable. The residual in b) does not visibly oscillate along the time axis. Also, it does not mask or alter the 250 fs oscillation observed in a), picked up with a Gaussian model.

The referee also states:

'I therefore remain unconvinced that the simple fitting procedure used to extract the shifts is doing a reliable job. There are clear systematic differences in the peak shape used in the fit compared to what is observed such that I do not believe the gaussian centre and width provides an accurate description of the changes occurring.'

We stated in the main text of the manuscript: 'The shifts are visible already in the raw data in Fig. 3 a) as described above, nevertheless we employ a fit to reduce the complexity of the raw data. The simplest fitting model to pick up the spectral fluctuations is a double Gaussian fit of the whole difference spectrum.'

All arguments in the paper used in the discussion of the oscillating signal in Fig. 3 are based solely on the raw data. We described the oscillatory feature even before introducing the fit (page 10 in the manuscript). None of the arguments brought forth in the discussion refers to the fit. We included this fit as a line to guide the eye, and we can remove it without changing any text in the discussion section.

As written above, we now completely cancel any mention of the Gaussian fit from the main manuscript and also remove it from the supporting information.

This leads to the following changes in the main manuscript:

Figure 3 a and b are now not containing the results of the Gaussian fit. The oscillation in the signal in Fig. 3a is well visible in false-color and contour lines. The comparison to theoretical predictions in Fig. 3b is equally valid without plotting the fit.

The caption now reads:

Figure 3: Experimental shifts and theoretical predictions on state population. a) False-colour contour plot of the positive lobe in Fig. 2b, normalized on the time-dependent area under the lobe. An oscillatory dynamics in the lineshape and position is visible for the first ~ 600 fs. At 150 fs and 400 fs delay, the spectrum is shifted to lower kinetic energy, while it is shifted to higher kinetic energies in between and afterwards. b) Comparison of the oscillation dynamics with trajectory simulations. The population of the S_1 ($n\pi^*$) state, obtained from CASPT2 calculations of Ref. ³⁵ (blue line) and ADC(2) calculations of Ref. ³⁶ (orange line, both on the top scale) are plotted. The dashed lines highlight the extrema of the oscillation observed in the experiment. The theoretical simulations do not include finite time-resolution and we shifted them by 50 fs to smaller delays to induce a transient rise of the signal around zero delay. The experimental 250 fs oscillation features have their counterparts in the simulated S_1 population, indicating the observation of a population exchange between the S_1 state and other electronic states.

The few sentences in the previous version referring to the Gaussian model in the supporting information are cancelled: 'The shifts are visible already in the raw data in Fig. 3 as described above, nevertheless we employ a fit to reduce the complexity of the raw data. The simplest fitting model able to pick up the spectral fluctuations is a double Gaussian fit of the whole difference spectrum. While it might not be the best fit function, it is the simplest to quantify the spectral trends. Figure 3a displays the spectral position as well as the mean error as black symbols, with the 250 fs oscillation period clearly visible. More information about the fit is given in the Supplementary Discussion 1.'

The Gaussian model is also cancelled from the supporting discussion 1.

We included in the supporting information a new supplementary discussion 3 showing two lineouts of Fig. 3 – one at the right and one at the left side of the maximum. We refer to this graph from the main text in the paragraph below fig. 3.

3 Spectral oscillations

To reduce the complexity of the data shown in Fig. 3, we show lineouts at the positive lobe's low and high kinetic energy edges at 98.8 eV (blue) and 101.2 eV (orange) in Fig. S2. In the delay region described above, both lineouts show an oscillation. They occur with opposite phase as expected for spectral shifts. At 150 and 400 fs, the underlying photoelectron signal shifts to lower kinetic energies, increasing the lineout at

98.8 eV while decreasing the lineout at 101.2 eV. In between these times, the photoelectron signal shifts to higher kinetic energies, reducing the lineout at 98.8 eV and increasing the 101.2 eV lineout. At higher delays, the step size is too coarse to resolve such oscillation.

Figure S2: Lineouts for the delay-dependent difference signal at 98.8 eV (blue line) and 101.2 eV (orange line). The two curves show an out-of-phase oscillation within the first 600fs. The grey dashed vertical lines mark the same positions as the dashed lines in Fig. 3.

Please find attached the manuscript with changes indicated in yellow. We hope that the paper in its current form meets the standards for publication in Nature Communications. We thank you for your patience!

Best regards,
Markus

REVIEWERS' COMMENTS

Reviewer #2 (Remarks to the Author):

The new figure certainly shows the lack of oscillations in the residuals. And the new SI figure highlights the oscillation more clearly. I can't say I am completely convinced by the interpretation but it does provide a reasonable explanation of the data as shown. I am therefore happy for the manuscript to be accepted.

I don't need to see the manuscript again but it would be good if the authors could add a discussion of how the energy corrections (detailed in section 9 of the SI) and shifts of the photoline affects the appearance of the oscillations? For example how large are the shot to shot variations of the energy between the UV on and UV off measurements?

Reply to referee's comment

We would like to thank the referee for the comments.

Referee #2:

The new figure certainly shows the lack of oscillations in the residuals. And the new SI figure highlights the oscillation more clearly. I can't say I am completely convinced by the interpretation but it does provide a reasonable explanation of the data as shown. I am therefore happy for the manuscript to be accepted.

We are happy that we could convince the referee on the existence of the observed oscillations.

I don't need to see the manuscript again but it would be good if the authors could add a discussion of how the energy corrections (detailed in section 9 of the SI) and shifts of the photoline affects the appearance of the oscillations? For example how large are the shot to shot variations of the energy between the UV on and UV off measurements?

We have extended the Supplementary Discussion 9 and we now briefly discuss two jitter sources (photon energy and temporal jitter) that affect the difference spectra and how the corrections alter the spectra. The following text has been added:

“Influence of the data handling on the experimental spectra. Both the spectral fluctuations of the x-ray pulse and the pump-probe delay fluctuations are potentially smearing out spectral-temporal signatures of the molecule. These fluctuations are resulting from the fact that the free-electron-laser is starting its lasing process from noise, part of the temporal jitter is due to other stabilization issues. Several papers have been devoted to investigate the issues of delay jitter (in Suppl. Refs. ^{8,9}) and we systematically investigated the effect of spectral and timing jitter on time-resolved photoelectron spectra in Suppl. Ref. ¹⁰.

The delay time jitter in our data is about 300 fs. While the photon energy jitter primarily changes the position and the width of the photoline, this delay jitter may also change the shape of the observed photoline. Applying a correction to the delays by tracing the arrival time of the x-ray pulses already eliminates a significant part of the random fluctuations in the time-dependent difference spectra. This has been demonstrated before in Suppl. Refs. ^{8,9}. For the difference spectra here, it makes the spectral oscillations in the region between 100 and 101 eV become visible (see Figure 4b in Suppl. Ref ¹⁰).

The most obvious ‘missing’ spectral feature is the sulphur photoelectron line spin-orbit splitting of 1.2 eV, that cannot be identified in our data. Since we do not have a single-shot spectral tool available, we used self-referencing for spectral correction. Based on a combination of simulations and data, we found that shot-to-shot correction by self-referencing of the photoline can correct the jitter and drifts in the static case. For pump-probe difference spectra, however, the correction of unpumped shots can only be achieved by well-adapted averaging of the data and utilising correlations in the pulse train of the FEL. Nonetheless, this allows to correct long-term drifts of the FEL photon energy. Thus, our method does not influence the shot-to-shot statistics, but narrows the long-term averaged statistics and improves it by a factor of 2-3.” Further the following Reference has been added to the Supplementary Reference list:

“10. Mayer, D., Lever, F. & Gühr, M. Data analysis procedures for time-resolved x-ray photoelectron spectroscopy at a SASE free-electron-laser. *J. Phys. B At. Mol. Opt. Phys.* (2021) in press doi:10.1088/1361-6455/ac3c91.”